# ModSoft-HP: Fuzzy Microservices Placement in Kubernetes

Euripides G. M. Petrakis *,† , Vasileios Skevakis †, Panayiotis Eliades †, Alkiviadis Aznavouridis †
and Konstantinos Tsakos †

School of Electrical and Computer Engineering, Technical University of Crete (TUC), 73100 Chania, Crete, Greece;
vskevakis@tuc.gr (V.S.); piliadis@tuc.gr (P.E.); aaznavouridis@tuc.gr (A.A.); ktsakos@tuc.gr (K.T.)
* Correspondence: epetrakis@tuc.gr
† These authors contributed equally to this work.

**Abstract:** The growing popularity of microservices architectures generated the need for tools that orchestrate their deployment in containerized infrastructures, such as Kubernetes. Microservices running in separate containers are packed in pods and placed in virtual machines (nodes). For applications with multiple communicating microservices, the decision of which services should be placed in the same node has a certain impact on both the running time and the operation cost of an application. The default Kubernetes scheduler is not optimal in that case. In this work, the service placement problem is treated as graph clustering. An application is modeled using a graph with nodes and edges representing communicating microservices. Graph clustering partitions the graph into clusters of microservices with high-affinity rates. Then, the microservices of each cluster are placed in the same Kubernetes node. A class of methods resorts to hard clustering (i.e., each microservice is placed in exactly one node). We advocate that graph clustering should be fuzzy to allow high-utilized microservices to run in more than one instance (i.e., pods) in different nodes. ModSoft-HP Scheduler is a custom Kubernetes scheduler that takes scheduling decisions based on the results of the ModSoft fuzzy clustering method followed by heuristic packing (HP). For proof of concept, the workloads of two applications (i.e., an e-commerce application, eShop, and an IoT architecture) are given as input to the default Kubernetes Scheduler, the Bisecting K-means, and the Heuristic First Fit (hard) clustering schedulers and to the ModSoft-HP fuzzy clustering method. The experimental results demonstrate that ModSoft-HP can achieve up to 90% reduction of egress traffic, up to 20% savings in response time, and up to 25% less hosting costs compared to service placement with the default Kubernetes Scheduler in the Google Kubernetes Engine.

**Keywords:** service-oriented architecture; Kubernetes Scheduler; fuzzy clustering; modularity optimization

## 1. Introduction

Kubernetes (K8s) automates the deployment and orchestration of containerized applications across server infrastructures and in the cloud. Applications run in clusters of nodes. The services are placed in nodes in a way that minimizes any of several criteria (e.g., latency, number of nodes, operation costs, security, fault tolerance) or a combination of them. Configuring a Kubernetes cluster resorts to Kubernetes Scheduler [1]. The decision to place a service in a node is taken based on (available and required) resources and user preferences. The default Kubernetes Scheduler can be extended to address the requirements of service deployment in heterogeneous or federated infrastructures [2,3]. For example, service deployment in fog-edge computing environments dictates the placement of services closer to network edges [4].

An assumption common to all these works is that the workload comprises independent services (monolithic applications). However, the default scheduler cannot take placement decisions based on service dependencies (affinities). The problem often occurs

when deploying service-oriented-architecture (SOA) applications comprising multiple communicating services which do not all fit together in one node. The services must be placed in nodes in a way that minimizes the number of nodes and inter-node communication (i.e., egress traffic). We advocate that the decision of which services should be grouped and placed in the same node has an impact on both the running time (i.e., latency) and the operational cost of an application (i.e., cloud providers charge their clients based on the number of reserved machines, the resources allocated to each machine, and the network traffic between different machines).

In our recent work [5], all these objectives are met together by grouping the application microservices into smaller groups with high-affinity rates and by placing each such group within the same node. The problem of microservices placement is formulated as graph clustering (or graph partitioning). Both nodes and edges of the graph are labeled by the resources consumed (i.e., CPU and RAM) and by the affinities between microservices (i.e., network traffic), respectively. Clustering is hard (i.e., each graph node may belong to exactly one partition). Services that exchange high traffic rates are placed in the same node. Placing these services in the same node reduces the traffic exchanged between the infrastructure's nodes (i.e., egress traffic). In [5], several graph partitioning algorithms are tested and their placement solutions are compared against the solution of the default Kubernetes Scheduler in the Google Kubernetes Engine (GKE). Our proposed Bisecting K-Means (BKM) and Heuristic First Fit (HFF) [6] proved to be more efficient by optimizing almost all cost factors (i.e., number of nodes, egress traffic, infrastructure hosting cost). However, the hard partitioning approach did not reduce the applications' latency (response time) in all cases.

The performance of a microservice placement solution can be greatly improved by applying [7] fuzzy clustering (rather than hard clustering) to partition the application graph. To the best of our knowledge, fuzzy clustering has not been exploited for the problem of service placement elsewhere in the literature. Fuzzy placement solutions allow more than one instance (replica) of certain services to be placed in different nodes. This reduces the workload of each microservice replica by load-balancing the incoming requests. This results in faster application response times without impacting the hosting cost.

Modularity soft (ModSoft) clustering [8] followed by heuristic packing (HP) [9] leads to a fuzzy service placement solution embedded into a custom Kubernetes (K8s) Scheduler. Application deployment comes along with appropriate supporting services that produce the graph of an application, *a service mesh* [10], with monitoring services and a middleware service that automates service placement to GKE (i.e., translates the output of graph clustering to Docker Compose). We present comparisons of the proposed fuzzy method with placement with Bisecting K-Means [5], Heuristic First Fit [6], and the default Kubernetes Scheduler [1] based on several performance metrics (i.e., response time, infrastructure resources, and hosting cost). For proof of concept, service placement results from all methods are reported for two real-world applications (i.e., the Google Online Boutique eShop demo application [11] and iXen IoT architecture [12].

Section 2 presents work related to the scheduling of applications with multiple communicating services. The Kubernetes configuration for scheduling service-oriented architecture applications is discussed in Section 3. Kubernetes is enriched with additional services to support the proposed (user-defined) fuzzy service placement solution. Fuzzy clustering and the proposed fuzzy placement algorithm are described in Section 4. Section 5 presents the evaluation methods and the Kubernetes testbed. Affinity metrics and evaluation comparison criteria are also discussed. Section 6 reports experimental results using the default Kubernetes Scheduler of GKE, two hard-clustering algorithms, and ModSoft-HP. Conclusions and issues for future work are discussed in Section 7.

## 2. Related Work

Service placement is a well-known problem and has been studied extensively in the literature [13,14]. The ever-increasing popularity of containerized applications has generated additional interest in scheduling solutions that run on Kubernetes (K8s) [2,3].

Existing methods are categorized by type of infrastructure (i.e., single or heterogeneous clouds or cloud-fog) and by optimization policy. The emphasis is on placing independent monolithic applications (i.e., each service is an application) on the servers of a platform.

In Kubernetes and the cloud, service placement is determined by the scheduler(s) of the provider. These are heuristic algorithmic solutions that pair the resource demands of each service (e.g., CPU, RAM) and user preferences with the resource constraints of a server (or VM) to decide its placement. Often, these servers belong to clusters in the same cloud. To guarantee high availability, the cloud providers (or the developers themselves) may choose to place the services in different zones, regions, or different clouds (i.e., federated clouds). In that case, the end-users are opted to tolerate additional service charges and communication delays for services deployed in different server clusters. The cloud-fog (or the cloud-edge) environment, in particular, calls for solutions that place latency-critical services closer to their end users [15]. In general, solutions to the service placement problem must preserve acceptable quality of service (QoS) level subject to (possibly several) application constraints (e.g., security, cost, latency, fault-tolerance, and general service level agreements). Typically, the scheduler runs in the master node and schedules only one application (or service) at a time. Distributed or self-adaptive scheduling solutions have also been proposed [4].

The advent of SOA architectures [16] generated the need for advanced coordination and management of resources (i.e., service deployment, monitoring, scaling). At the very basic level, Docker facilitates the build and management of SOA applications with their services deployed as containers. At a larger scale (e.g., at the cloud provider level), Kubernetes is a prominent platform for managing and orchestrating multiple containerized applications. Kubernetes provides a framework to run distributed systems resiliently by offering service discovery, load balancing, storage orchestration, scalability, automated service placement, self-healing of containers, and enhanced security. SOA scheduling in Kubernetes must consider (in addition to the resource demands of each service) the traffic (i.e., *affinities*) between collaborating services. This is a relatively new problem and has not been studied in depth in the literature. The assignment of services to servers resorts to the default *Kubernetes Scheduler* [1], which has been proven to be sub-optimal for SOA [5].

The following is a review of service scheduling solutions for SOA in the cloud and cloud-fog infrastructures, including solutions for Kubernetes. Their emphasis is on minimizing the number of hosts (VMs or nodes in K8s), an affinity metric, latency, infrastructure costs, or all of them combined. Most methods model an application by means of a graph. Graph nodes represent services (and their resource demands) and graph edges represent their message traffic (e.g., requests per second). A placement solution is determined by partitioning the graph into clusters of nodes with high affinity. All works report results based on simulations and mock-up SOA applications (rather than on real-world applications and real workloads). Only a few of them are tested on real server platforms (in the example of ModSoft-HP).

All methods rely on the idea that each service belongs to exactly one node. ModSoft-HP relaxes the requirement of each service belonging to exactly one node and proves that this results in a better placement solution compared to existing methods (e.g., [17,18]) and our previous work [5]. Also, ModSoft-HP and [19] adapt the initial placement to the resource demands of floating (i.e., changing) workloads. Table 1 summarizes the results of this review.

**Table 1.** Comparison table.

| Method/Features | Model | Platform | Application | Placement |
|---|---|---|---|---|
| ModSoft-HP | graph (fyzzy clustering) | Google Kubernetes Engine | eShop, iXen | adaptive |
| [5] | graph (hard clustering) | Google Kubernetes Engine | eShop, iXen | static |
| [18] | graph (K-cut partitioning) | Exogeni simulator | simulation | static |
| [17] | graph (K-cut partitioning) | Amazon EC2, CloudSim | simulation | static |
| [6] | REMAP (heuristic) | Kuberbetes on Azure VMs | Sock-shop | static |
| [19] | HTAS (heuristic) | Kuberbetes on Nectar VMs | simulation | adaptive |
| [20] | graph (affinity heuristic) | cloud simulator | simulation | static |
| [21] | graph (cost flow) | Exogeni simulator | Google traces | static |
| [15] | optimization (heuristic) | iFogSim simulator | simulation | static |
| [22] | optimization (heuristic) | edge-cloud simulator | simulation | static |
| [23] | optimization (heuristic) | edge-cloud simulator | simulation | static |

Recent approaches model the problem of placing microservices in Kubernetes as a graph partitioning one. Hu, Laat, and Zhao [18] run several experiments using synthetic workloads on the Exogeni [24] cloud simulator. Sampaio et al. [6] proposed REMaP, a service placement for Kubernetes that proved to improve the performance and reduce the number of K8s nodes of an application. They implemented a monitoring component using Influxdb [25] and Zipkin [26] and used mock-up and empirical evaluations on artificial service topologies to assess the performance of their method.

Huang and Shen [17] proposed a service deployment method to reduce application response times rather than the cost of application hosting. They modeled an application using graphs that represent the communication costs and the parallelism between services. They handled the problem of service placement in VMs as a minimum k-cut problem. They showed performance results for four service deployment methods on Amazon EC2 and CloudSim simulator [27]. They did not use Kubernetes and run all experiments on only one VM. Bhahmare et al. [20] dealt with the problem of scheduling microservices on different types of cloud environments. They showed reduction in the communication of microservices and improved response times to requests. Hu and C. Laat and Z. Zhao [18] modeled service placement as a graph partitioning problem and used service affinities to re-arrange services into the nodes. Both methods rely on simulation results and have not been tested in a realistic environment and in real use cases. Zhong and R. Buyya [19] proposed a task allocation strategy for Kubernetes in a heterogeneous environment. Their method relies on a sufficient job configuration policy, cluster size adjustment, and a service re-scheduling mechanism that led to the cost reduction in application hosting by reducing the number of nodes. Hu at al. [21] modeled K8s scheduling as a cost flow problem on the service graph and provided a realistic evaluation of performance using data and a large Google cluster trace on the Exogeni simulator.

In a recent work [5], we solve the service placement problem from the scope of reducing the total (monetary) cost of application hosting. An application is modeled by means of a weighted and directed acyclic graph with two different affinity metrics, one counting the number of requests between services and also one counting the message size (in bytes) exchanged between the services. Several graph clustering algorithms are implemented and evaluated. Among them, the Heuristic First Fit (HFF) [6] and our proposed Bisecting K-means (BKM) in combination with heuristic packing (HP) [9] produced the best results. Heuristic packing is a post-processing step that determines if there is a service placement solution with even fewer nodes. All algorithms limit each service to a single instance placed on a node (i.e., hard clustering). All methods outperformed the service placement solution of the default Kubernetes Scheduler in all criteria (i.e., number of nodes, egress traffic, application hosting cost).

Farhadi et al. [22] presented a two-scale framework for joint service placement and request scheduling in edge clouds for data-intensive applications. They used simulations for testing. Pallewatta, Kostakos, and Buyya [15] proposed a decentralized microservice placement policy for heterogeneous and resource-constrained fog environments. Each microservice is placed at the nearest data center to minimize latency and network usage. Their method improved latency and reduced the delay in network communication. Apat, Sahoo, and Maiti [23] proposed a service placement model for minimizing the energy consumption in a fog environment. They did not test their model in an actual fog environment.

## 3. Kubernetes Configuration

A Kubernetes (K8s) cluster is a collection of *node pools* (i.e., groups of nodes with the same configuration). Each node runs a container-optimized operating system (OS) and hosts several pods. As a matter of good practice, each microservice is placed in a separate pod (it is easier to debug, troubleshoot, or inspect the services). A node pool is configured with the CPU, RAM, storage space, and OS requirements that each initialized node must meet according to each application's requirements. Node pools with a different configuration can be added as well. A Kubernetes cluster includes at least one *worker node* and a *master node* (or *control plane*) which host all Kubernetes components.

### 3.1. Network Traffic

Kubernetes uses IP addresses to enable communication between pods (microservices) and K8s components. A pod is assigned an IP address upon creation. This IP address is temporary (ephemeral) and changes every time the pod restarts (i.e., due to a crash or update). For this reason, Kubernetes introduced a resource called *Kubernetes Services*. Kubernetes Services are abstractions (configurations) that allow the pod to use the network to communicate safely (either internal cluster communication or external network communication). Each pod bounds with its respective Kubernetes service that is responsible for forwarding any traffic to the pod. The service discovers the pod's IP address upon creation (or change) and exposes a permanent address (user-defined in the Kubernetes Services YAML configuration) and a port so that other services can communicate with it.

The Kubernetes Services are network configurations and not service instances. They offer an address in order for a request to be forwarded to their respective pod (microservice). Each Kubernetes service is assigned a unique IP address (clusterIP). This address is tied to the lifespan of the service and will not change while the service is alive. Kubernetes Services act as load balancers for oncoming traffic if multiple pod instances are attached to the service. If there is a replica set of pods, Kubernetes Services will choose the *optimal* pod to forward each request. In this work, we assume that Kubernetes Services will choose to forward the requests to a destination pod deployed in the same node as the pod issuing the request (if such a pod exists).

Figure 1 shows an example application deployed in nodes A and B. The blue box inside each node represents the *kube-proxy* component that contains the configurations of the Kubernetes Services (Service A, Service B, Service C) and has access to all Kubernetes Services communication information (addresses). The Service C pod (instance of Service C) delivers the request to Kubernetes Service A, which then forwards the request to the Service A pod (optimal pod), located in the same node as Service C and not to the Service A pod in node B. Gray lines in the figure represent *ingress* traffic, while blue lines represent egress traffic. In this work, if one of the destination pod instances is located in the same node as the source pod, the traffic between these pods is considered as ingress traffic (in-node traffic).

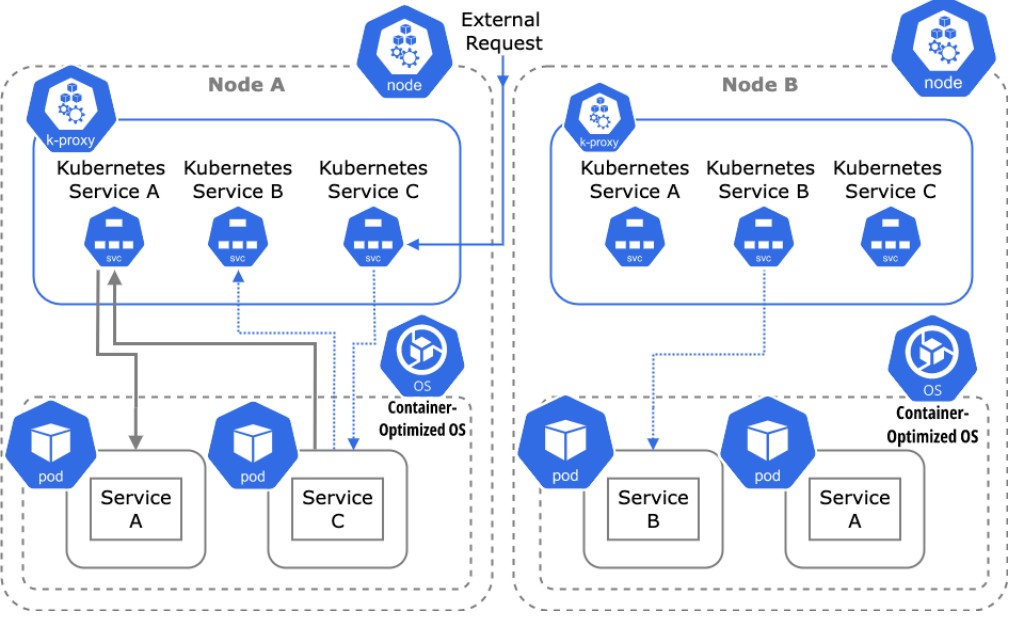

**Figure 1.** Traffic between pods.

### 3.2. Service Mesh and Service-Oriented Architectures

Microservices in Kubernetes must be configured to discover where other microservices are deployed. The developers of the service-oriented architecture must implement additional logic to handle communication, security configuration, and fail logic. A *service mesh* [10] is

a dedicated infrastructure layer for applications to add capabilities for observability, traffic management, and secure communication. The control plane of the service mesh injects a *sidecar proxy* service as a third-party application, which handles all the logic mentioned above.

Figure 2 illustrates the architecture of a pod with service mesh. Without the service mesh, the developer must implement (in addition to the microservice logic) a communication configuration, security configuration and fail logic, and metrics, as shown in the middle figure; this logic is packed within a sidecar proxy (figure in the middle), and it is automatically configured and deployed within the Envoy sidecar proxy as illustrated in the figure on the right.

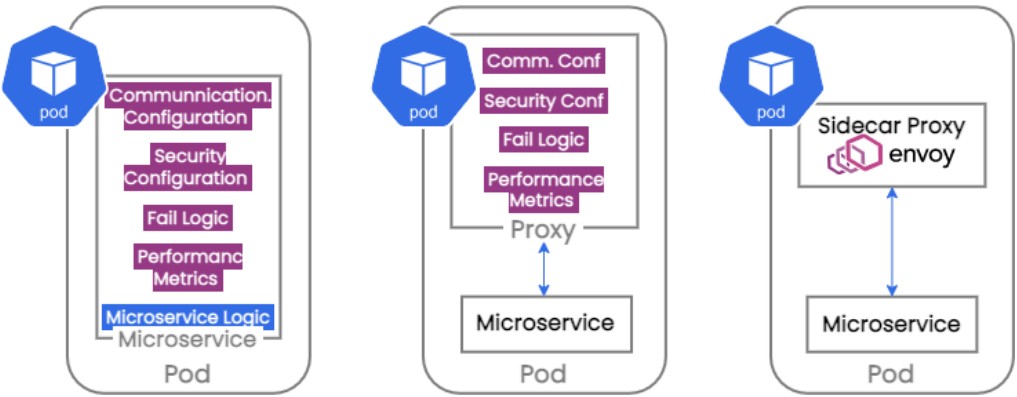

**Figure 2.** Pod architecture.

*Istio* [28] is an open-source implementation of a service mesh, and *Istio* is the control plane of Istio. Istio automatically detects new services and endpoints in the cluster and deploys an *Envoy sidecar proxy* service in each newly created pod. Istio also manages all the certificates and configures secure TLS communication between the services. Finally, Istio collects telemetry data and exports metrics from each pod, which can then be acquired by a monitoring server, like *Prometheus* [29]. Many services do not have native Prometheus support, so an extra component, called an *exporter*, is deployed to read the metrics from the service, transform them into a compatible format, and expose the endpoint so Prometheus can pull these metrics. In addition, Kiali [30] is a management console for Istio and provides a powerful way to visualize the topology of the service mesh by creating the Kiali graph, which displays the services' network communication protocol, their traffic rates, and the latency between them. Kiali is deployed as a service, and it offers an API through which the mesh information and the Kiali graph can be obtained.

### 3.3. Kubernetes Cluster

The Kubernetes cluster in Figure 3 is deployed in the Google Cloud Platform (GCP). Details on the customization of the Kubernetes cluster can be found in [31]. Istio is randomly deployed in one node and configured to communicate with the service mesh created by the Envoy sidecar proxies to log all network traffic on the cluster. A Prometheus node exporter is deployed in every node and is responsible for exporting node metrics so that the Prometheus server can pull and store these metrics in real time. The Prometheus server is deployed randomly among the available nodes (based on the scheduler's decision). The Kiali service is also randomly deployed among the available nodes and is configured to pull Istio's logged network traffic through the Prometheus server.

Each Kubernetes node can host a finite number of pods (microservices), depending on the available node resources. Each microservice has a respective associate Kubernetes service responsible for all microservice communication. Suppose a microservice is deployed as a *replica set* (i.e., there will be more than one instance of the microservice in separate pods). In that case, the associate Kubernetes service is responsible for forwarding the requests between the instances, as mentioned in Section 3.1. All the traffic within the pods is forwarded through the injected Envoy sidecar proxies. Envoy proxies are responsible for receiving incoming and outgoing requests to the pod's microservice. If the request is

incoming, the proxy forwards it to the microservice through the respective Kubernetes service. If the request is outgoing, it is sent to the targeted pod so it can be processed from that pod's Envoy proxy. Istio monitors the service mesh, which is the communication between the Envoy sidecar proxies. Every pod contains exactly one microservice, described as a Kubernetes deployment (configured in a YAML file).

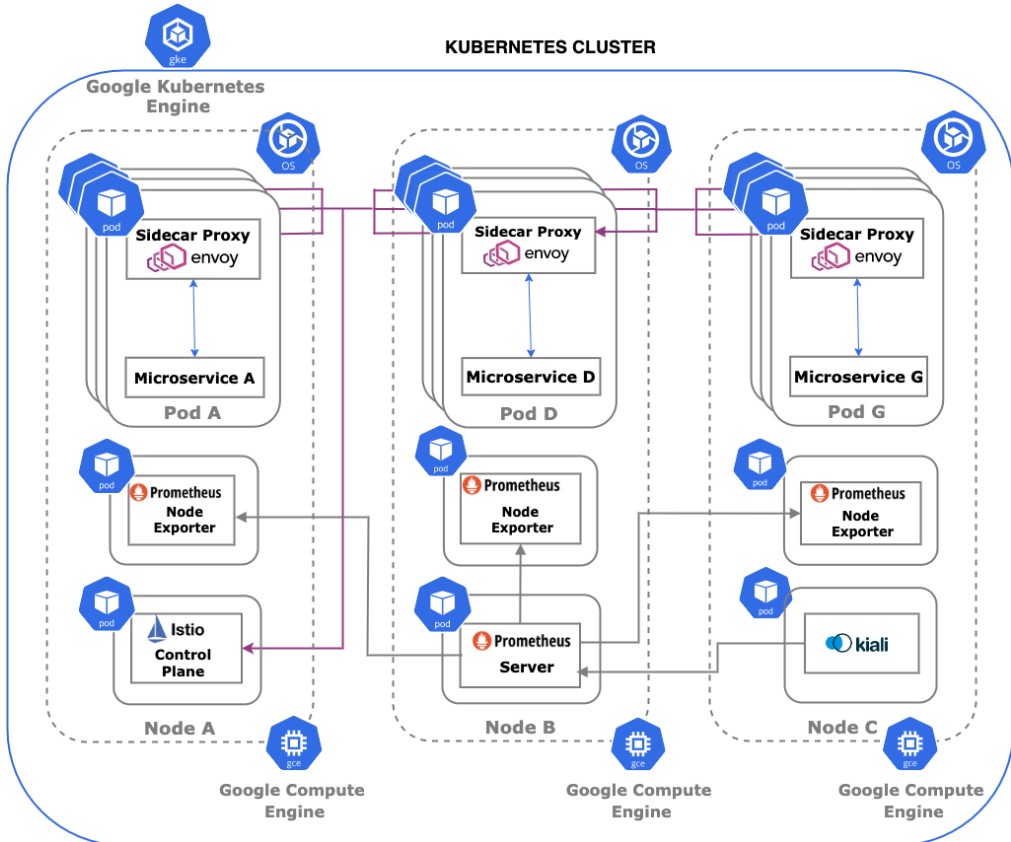

**Figure 3.** Kubernetes cluster deployed in the Google Cloud Platform.

## 4. Fuzzy Microservices Placement

The output of hard clustering algorithms is a partition of the dataset where each element belongs to exactly one cluster. Fuzzy (or soft) clustering computes the probability of each element belonging to a cluster [7]. For example, fuzzy C-Means (FCM) [32] is a fuzzy extension of the standard K-means algorithm. Modularity optimization [33] detects node clusters (modules) in graphs based on *modularity*. Modularity is defined as the difference between the probability of two nodes of a graph belonging to a partition and the probability of two random nodes belonging to the same partition. It measures the *quality* of a module as the density of the connections within a module. The modularity score of a node may increase (or decrease) by including one of its neighbors in its community. The Louvain algorithm [34] is a popular implementation for modularity optimization. It is widely used in community detection in large networks. Its application in microservices placement has not been investigated in the literature. Perhaps this is due to the following two reasons: first, it is inefficient for large graphs (i.e., modularity optimization is NP-hard) and, second, it cannot detect small communities (e.g., graph partitions with only a few nodes). In regards to complexity, several polynomial time approximations to the problem of modularity optimization are known to exist [35]; in regards to the second issue, fuzzy approaches seem to be more effective.

*4.1. ModSoft Algorithm*

Modularity optimization produces a hard clustering solution. Soft modularity optimization [36,37] produces fuzzy partitions by calculating a membership matrix $p \in R^{n \times K}$, where $p_{ik} \geq 0$ is the degree of membership of node $i \in V$ to the $k$-th cluster. $V$ is the set of graph nodes, and $K \in [1, K]$ is the maximum number of clusters. *ModSoft* [38] is an efficient approximate solution to the soft modularity problem. ModSoft introduces a membership matrix $p \in R^{n \times n}$ which is independent of the number of clusters $K$ (i.e., the number of clusters need not be known in advance). Node memberships to a cluster depend on its neighbors and do not require the processing of the entire graph. The worst case complexity of ModSoft is $\mathcal{O}(n^2)$, where $n$ is the number of nodes. Most nodes produce zero membership probabilities which facilitate graph storage, detection, and processing of graph partitions.

Algorithm 1 is an adaptation (for microservices placement) of the reference implementation of ModSoft [8]. $W$ is the adjacency matrix of the graph ($W_{i,j}$ is the weight of edge $(i, j)$), $w_{i,j} = \sum_{j \in V} W_{i,j}$ is the weighted degree of a node, $w = \sum_{i \in V} w_i$ is the total weight of the graph. The probability that an edge exists is $w_i \cdot w_j / w$. The algorithm works in three stages:

*Initialization of membership matrix (lines 4–9)* The *membership matrix $p$ ($n \times n$ matrix) is initialized to the one's matrix ($I$) and the *weighted average vector $\overline{p}$* is initialized to the weighted degree of each node (i.e., sum of the weights of the node's edges). Each row of the matrix represents the probability distribution for a node to belong to a partition, which sums to 1.

*Calculation of membership matrix (lines 10–15)* The membership matrix $p$ is updated at each step of an iterative process. At the end of each iteration, the modularity is calculated, evaluating the optimality of the partitions. This process is repeated until the increase in modularity falls below a given threshold *MT*, which we set at 0.01. The final result is the membership matrix. Each cell of the membership matrix represents the probability of the row node being in the same partition as the column–row node (which is a value in $[0, 1]$). The sum of the probabilities of each row equals 1. During each iteration, a gradient descent step is performed as an update rule for the membership of each node. The gradient descent step is calculated locally (i.e., only the neighbors of each node are used to update the node's membership) and aims to maximize the modularity. After the gradient descent step, a projection is performed and the (average) vector $p$ is updated. The *update_membership* step includes both the membership update function and the projection step and it returns the updated membership matrix $p$.

*Partitioning (lines 16–26)* The final step iterates over the final membership matrix $p$ to calculate the microservices partitions. As mentioned before, the membership matrix rows and columns represent the application services, and the value of each cell is a number between 0 and 1 which represents the probability of the row service being in the same partition as the column service. We create a partition for each row service by using a predefined threshold $t$, which we set at 0.1. We place each column service whose value is over our threshold in the partition of the row service. Finally, the partitions are sorted by the number of services they contain. The algorithm produces the fuzzy partitions $P$. The fuzziness parameter $t$ controls the fuzziness of the produced partitions. A higher value of $t$ results in less-fuzzy partitions and the reverse (i.e., a high value might produce fewer fuzzy partitions).

ModSoft can be applied directly to the application's graph. It relaxes the requirement of only one instance per service in Kubernetes nodes and allows more than one replica of each service to be placed in different clusters. Detail on the implementation of the ModSoft algorithm for microservice placement can be found in [31].

---

**Algorithm 1** Fuzzy Partitioning Algorithm

---

1: **procedure** MODSOFT(Graph, *P* partitions)
2:    **Input:** Services Graph (G), Graph nodes (V), Application Services (S), Threshold (T), Modularity Threshold (MT), Fuzzyness Parameter (t)
3:    **Output:** Microservices Partitions (*P*)
                                                     ▷ Initialize membership matrix *p*
4:    Calculate the total weight of graph G, total_weight
5:    Calculate the weighted degree of each node N, $degree_{node}$
6:    $i \leftarrow 1$
7:    **for** node $i \in$ V **do**
8:       $p_i \leftarrow 1$
                                                     ▷ Update membership matrix *p*
9:    $i \leftarrow 1$
10:    **repeat**
11:       $i \leftarrow i + 1$
12:       $p \leftarrow update\_membership(p, t)$
13:       $modularity_i \leftarrow modularity(p)$
14:    **until** $modularity_i - modularity_{i-1} \leq$ MT
                                                 ▷ Calculate services partitions *P*
15:    $i \leftarrow 1$
16:    **for** service(S) $i \in p$ **do**
17:       $P_k \leftarrow \{S_i\}$
18:       **for** service $j$ in $p_i$ **do**
19:          **if** $p_{ij} >$ T **then**
20:             $P_k \leftarrow P_k \bigcup \{S_j\}$
21:    Sort *P* by partitions with most services
22:    return *P*

---

### 4.2. ModSoft-HP Scheduler

In correspondence with the K8s scheduling problem, graph nodes are mapped to services, and graph partitions are mapped to K8s nodes. ModSoft produces partitions containing microservices. If the number of partitions is higher than the number of available nodes, heuristic packing (HP) [9] is applied to the output of ModSoft to determine if there is a placement solution with fewer nodes and to ensure that the services are optimally placed in the available cluster nodes using traffic rates, CPU utilization, and RAM utilization. The HP algorithm requires the fuzzy partitions, the node- and pod-requested resources, the list of services, and the list of affinities as inputs. It produces a cost-optimized solution (i.e., a placement solution that ensures that the least amount of resources will be allocated).

ModSoft-HP clustering decides where application services must be placed. The ModSoft-HP Scheduler invokes ModSoft-HP clustering to instruct the scheduler where (i.e., on which node) to place each service. The default Kubernetes Scheduler runs first to produce an initial placement. The initial placement is updated based on the results of an algorithm that partitions the graph of an application into clusters. The graph clustering algorithm runs either inside the cluster or remotely on a server (so that it does not waste cluster resources). The scheduler will automatically update the YAML of each service (deployment) to migrate the service to the new host. ModSoft-HP might dictate the replication of service in more than one pod. When a service is to migrate to a new node, Kubernetes will wait until the status of the new node turns to *RUNNING* before the service is terminated in the old node (so no downtime is observed until the new placement takes effect).

Algorithm 2 outlines this process. Details can be found in [39]. Line 4 will obtain the Application Default Credentials (ADCs). ADC is a strategy used by Google authentication libraries to automatically find credentials based on the application environment. Line 5 will get the cluster configuration from GCP using the project name, cluster name, and

cluster zone as input. Lines 6 will generate a *kubectl* configuration from the authentication of the cluster. Line 7 will create a cluster instance to run *kubectl* requests to the cluster and make changes in the deployment. Line 8 will get the initial placement using the default Kubernetes Scheduler. The name of the clustering algorithm to apply on the application's graph is selected in Line 9 and the algorithm is executed to produce the desired placement of pods to nodes (*finalPlacement*) in the cluster. If a service was on a different host in the initial placement, it is migrated to the new host. To migrate the service, the YAML file of the deployment is updated using *node affinity*, which constrains which nodes a pod can be scheduled on based on node labels. Additional detail about Kubernetes Scheduler customization can be found in [1–3].

---

**Algorithm 2** ModSoft-HP Kubernetes Scheduler

---

1: **procedure** MODSOFT-HP-SCHEDULER(Graph, *P* partitions)
2:　　**Input:** Project name, Cluster name, Cluster zone, Algorithm
3:　　**Output:** Placement of Pods to nodes (YAML file)
　　　　　　　　　　　　　　　　　　　　　　　▷ Authenticate and initialize cluster
4:　　Get the Application's Default Credentials (ADC)
5:　　Get the cluster configuration from GCP
6:　　Create *kubectl* configuration
7:　　Create cluster instance
　　　　　　　　　　　　　　　　　　　　　　　▷ Apply service placement logic
8:　　initialPlacement = $getInitialPlacement()$
9:　　SelectedAlgorithm = $chooseAlgorithm(Algorithm)$
10:　　finalPlacement = $executeAlgorithm(SelectedAlgorithm)$
11:　　**for** service ∈ finalPlacement **do**
12:　　　　**if** initialPlacement(service) ≠ finalPlacement(service) **then**
13:　　　　　　updateDeploymentYaml(service)

---

## 5. Evaluation and Testbed

The Kubernetes cluster is configured (Table 2) in the *europe-west3-b* region of GCP using the latest stable version of GKE (1.21.11). Horizontal pod autoscaling [40], along with all load balancing features of the GKE are enabled. This means that once the ModSoft-HP scheduling is applied, resources available to VMs will adapt to workload variations by adding new pods. Anthos service mesh [41] is disabled (at the time of the implementation it is in the beta phase). We chose to implement our service mesh using Istio. The Google Cloud Managed Service for Prometheus [42] is also disabled (it was in the beta phase on the GKE). We choose to apply the Prometheus tool in our cluster, which is stable and very well-documented.

**Table 2.** Cluster configuration.

| Cluster Attributes | Configuration |
| --- | --- |
| Location Type | Zonal |
| Zone | europe-west3-b |
| Release Channel | Regular |
| Cluster Version | 1.21.11-gke.1100 |
| Horizontal Autoscaling | Enabled |
| Vertical Autoscaling | Disabled |
| HTTP Load Balancing | Enabled |
| Managed Service for Prometheus | Disabled |
| Anthos Service Mesh | Disabled |

A node pool is configured for our cluster as shown in Table 3. The node pool contains all the specifications for the VMs that will be spawned as cluster nodes. The selected machines are of *e2-standard-2* type, each with two vCPUs, 8 GB of RAM, and a standard

boot disk with 40 GB available for storage. The VMs are located in the same zone as the cluster, and autoscaling is enabled. The OS of these machines is a Linux-based container-optimized OS. In [5], we showed that the benchmarking applications, deployed along with Istio, Kiali, and Prometheus, require at least two host machines (nodes) to run efficiently. For our experiments, we initialize four nodes for our clusters.

**Table 3.** Node Pool configuration.

| Node Pool Attributes | Configuration |
|---|---|
| Machine Type | e2-standard-2 |
| vCPU | 2 |
| RAM | 8 GB |
| Zone | europe-west3-b |
| Image | Container-Optimized OS with Container |
| Autoscaling | Enabled |
| Boot Disk | Standard/40 GB |

*5.1. Affinity Metrics*

Two affinity metrics are used to evaluate the communication between services and will be added as weights to the application graph. Choosing one or the other has a different impact on the estimation of performance and costs.

The *requests-per-second* (RPS) metric measures the requests forwarded from one service to another per second. RPS measurements are acquired from the Kiali API in JSON format. Kiali calculates RPS using Equation (1). $S_i$ is the source service, $S_j$ is the destination service, and $x$ is the total time of measurement in seconds. For TCP connections, Kiali reports only the size of messages sent or received (in bytes). To convert it to RPS, the size measures are combined with metrics from Istio and Prometheus.

$$\text{RPS}_{S_i \rightarrow S_j} = \frac{\text{Sum of Requests from } S_i \text{ to } S_j \text{ in } x \text{ seconds}}{x \text{ seconds}} \tag{1}$$

The *weighted bidirectional affinity* (WBA) [5] also exploits the size of the exchanged messages (in bytes) between two microservices as well as the total number of these messages. It is calculated according to Equation (2). $A_{a,b}$ is the affinity metric between service $a$ and service $b$, $m$ is the total number of messages exchanged, $m_{a,b}$ is the messages exchanged between $a$ and $b$, $d$ is the total amount of data exchanged in bytes, $d_{a,b}$ is the amount of data exchanged in bytes between service $a$ and service $b$, and $w$ is a weight ($\{w \in \mathbb{R} \mid 0 \le w \le 1\}$) that denotes the significance of each affinity components (i.e., size and messages count). In this work, $w = 0.5$ (i.e., the two components are equally significant).

$$A_{a,b} = w \cdot \frac{m_{a,b}}{m} + (1 - w) \cdot \frac{d_{a,b}}{d} \tag{2}$$

*5.2. Infrastructure Hosting Cost*

The application hosting cost on the Google Cloud Platform (GCP) is calculated according to GCP pricing documentation [43]. GCP charges its customers for the CPU and RAM allocation (the prices vary per region). Ingress network traffic is not charged, while egress traffic is charged based on the amount of data exchanged between nodes. Allocated storage space is also charged by GCP, but in our work, the volume of storage space used by the clusters is relatively low; hence, the cost of storage is negligible. Table 4 shows the price per resource for the machine type used in this work.

**Table 4.** GCP pricing for e2-standard machine type.

| Resource | Cost (USD) |
| --- | --- |
| Predefined vCPU | $0.028103/vCPU/h |
| Predefined RAM | $0.003766/GB/h |
| Ingress Traffic | $0 |
| Egress Traffic | $0.01/GB |

The total CPU and RAM charges for each machine are calculated according to Equation (3) and Equation (4), respectively, based on the resources allocated per hour. The CPU metric represents the virtual cores committed, while the RAM metric is the amount (in GB) of RAM allocated.

$$
\begin{aligned}
\text{Cost}_\text{CPU} &= 2\text{vCPU} \times \text{vCPU}_\text{cost} \times \text{hours} \\
&= 2 \times 0.028103 \times \text{hours} \\
&= 0.056206 \times \text{hours}
\end{aligned} \tag{3}
$$

$$
\begin{aligned}
\text{Cost}_\text{RAM} &= 8\text{GB} \times \text{RAM}_\text{cost} \times \text{hours} \\
&= 8 \times 0.003766 \times \text{hours} \\
&= 0.030128 \times \text{hours}
\end{aligned} \tag{4}
$$

The *network traffic* is charged for each *cluster*, based on the amount of traffic exchanged between nodes (i.e., egress traffic). It can be calculated by summing the *requested bytes* between services placed in different nodes. GCP charges for the size of messages in requests but not in responses. The location of each node is also a key factor in calculating egress traffic cost, but in our work, the nodes are all based on *europe-west3*, so all egress traffic (according to GCP) is charged the same. If $i$ and $j$ services in a node $N$, $t(i \rightarrow j)$, the requested bytes between services $i, j$, and $t_e$ the egress traffic, the network cost function can be expressed as:

$$
\begin{aligned}
\text{Cost}_\text{Traffic} &= \sum_i^N \sum_j^N t_e(i \rightarrow j) \times \text{cost}_\text{egress} \\
&= \sum_i^N \sum_j^N t_e(i \rightarrow j) \times 0.01
\end{aligned} \tag{5}
$$

where

$$
t_e(i \rightarrow j) = \begin{cases} t(i \rightarrow j), & \text{if } i, j \text{ in different nodes} \\ 0, & \text{otherwise} \end{cases} \tag{6}
$$

The total cluster cost function for $n$ nodes can be calculated as

$$
\begin{aligned}
\text{TotalCost} &= \text{TotalCost}_\text{CPU} + \text{TotalCost}_\text{RAM} + \text{TotalCost}_\text{Traffic} \\
&= n \times (\text{Cost}_\text{CPU} + \text{Cost}_\text{RAM}) + \text{TotalCost}_\text{Traffic} \\
&= n \times (0.086334 \times \text{hours}) + 0.01 \times \text{GB}_\text{egress}
\end{aligned} \tag{7}
$$

The total cluster cost for the initial placement, which utilizes four nodes, is:

$$
\text{TotalCost} = 0.345336 \times \text{hours} + 0.01 \times \text{GB}_\text{egress} \tag{8}
$$

*5.3. Benchmark Applications*

All service placement methods are evaluated on two benchmarking applications.

**Google Online Boutique** eShop [11] is a cloud-native microservices demo application that Google uses to demonstrate the use of technologies like Kubernetes, Istio, and the gRPC protocol. It is a Web-based e-commerce application consisting of 11 microservices

(12 with the Redis cache), where users can perform multiple e-commerce-related actions. The application is stable, and it uses five different coding languages and two of the leading service-to-service communication protocols (HTTP and gRPC), and it is implemented and optimized for use with the Google Kubernetes Engine as well as Istio. Figure 4 illustrates the architecture of the eShop application.

- *Frontend Service* (*Go*): Exposes an HTTP server that serves the website to the Web and generates session IDs for all users automatically.
- *Cart Service* (*C#*): Stores and retrieves the items users place on their shopping cart in a Redis cache database.
- *Product Catalog Service* (*Go*): Provides the list of products (read from a JSON file) and the ability to search and get individual products.
- *Currency Service* (*node.js*): Fetches real currency values from the European Central Bank and converts one money amount to another currency. It is the highest QPS (i.e., queries per second) service.
- *Payment Service* (*node.js*): Charges the user-provided credit card info (mock) with the payment amount and returns a transaction ID.
- *Shipping Service* (*Go*): Estimates shipping cost based on the shopping cart and ships items to the given address (mock).
- *Email Service* (*Python*): Sends user an order confirmation email (mock).
- *Checkout Service* (*Go*): Retrieves the user cart, prepares the order, and orchestrates the payment, shipping, and email notification.
- *Recommendation Service* (*Python*): Recommends products based on what the user placed in its cart.
- *Ad Service* (*Java*): Provides text ads based on given context words.
- *Load Generator Service* (*Python/ Locust*): Simulates application traffic by continuously sending requests imitating realistic user shopping flows to the frontend service.

**iXen IoT Platform.** iXen [12] is a prototype service architecture for the IoT. It supports the processing of information acquired by a network of devices. It has been developed as a microservice-based architecture with portable, independent microservices. iXen is a three-tier architecture design model, with each tier (layer) implementing unique logic for its respective targeted user group. The first-tier user group includes the infrastructure owners and the system administrators, which can install and connect devices in the infrastructure. The second-tier user group is the application developers, which can create subscriptions to sensors and create applications with these sensors. The final user group includes customers who subscribe to developer-created applications. Figure 5 illustrates iXen's architecture.

- *Web Service*: Provides a Web interface so the users can use the application
- *Keyrock Service*: Provides a REST API so that users can register, provides policies about user rights, and uses OAuth2 tokens to authorize users.
- *AuthZForce Service*: Describes respective user access rights using XACML.
- *PEP Proxy Services*: Provides a security mechanism for services offering a public interface. Every request to these public services is being forwarded through its respective PEP proxy and only requests from authorized users with access to the service are being forwarded to the service.
- *Querying Sensors Service*: Converts a custom query syntax to Mongo queries on the Mongo DB in order to search for a device based on location, model type, type of measurement, or the unit of the measurement.
- *Orion Context Broker Service*: Publish/Subscribe service that receives measurements from devices and makes this information available to other services and users based on their subscriptions.
- *Cygnus Service*: Accepts data streams in NGSI compliant and can store them on multiple types of databases.
- *Comet Service*: Reads Orion entities stored in a MongoDB and manages historical sensor data.

- *Mashup Service*: Responsible for creating developers' applications with the aid of Node-Red, an open-source flow-based programming tool for the IoT.
- *Load Generator Service*: Written in Python for Locust [44], it continuously applies distributed requests on the application's endpoints, simulating realistic user traffic and IoT devices' measurements/updates.

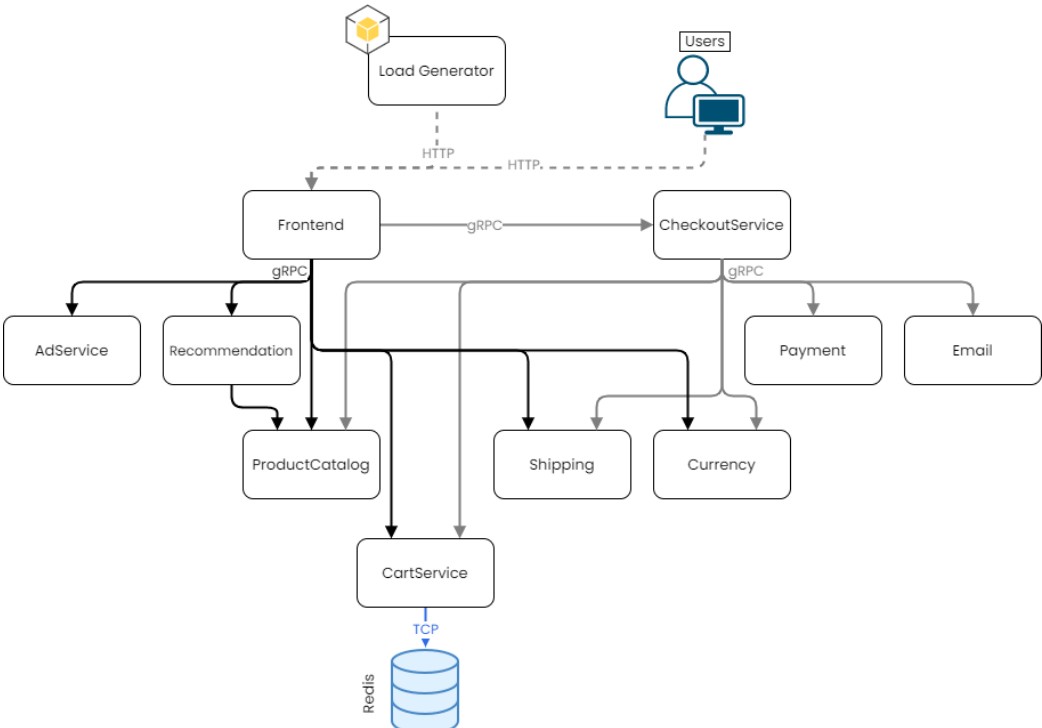

**Figure 4.** eShop architecture.

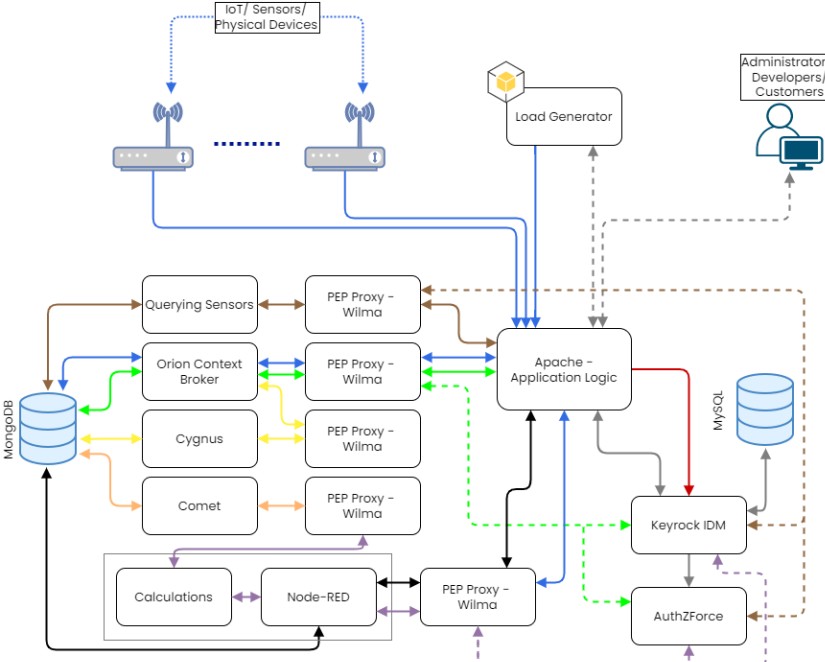

**Figure 5.** iXen Architecture.

### 5.4. Benchmark Application Stressing

For the stress testing of the applications, the *load generator* service is configured to issue requests (to the application endpoints of both applications) that simulate a load of concurrent users. These requests generate a traffic load through which we calculate the RPS and WBA affinity metrics to create the application graph.

For Google's online boutique eShop, the integrated load generator service is configured to simulate 10 users. We modified this service to simulate the load from 300 concurrent users using 57,706 requests with a (spawn) rate around 35 RPS. Table 5 shows the distribution of these requests by type.

**Table 5.** Stress-testing requests for the eShop application.

| Request | Request Type | Number of Requests | Distribution |
|---|---|---|---|
| Visit homepage | GET | 2498 | 4% |
| Show items in cart | GET | 7616 | 13% |
| Add item to cart | POST | 7632 | 13% |
| Submit an order | POST | 2538 | 4% |
| Obtain a product | GET | 32,396 | 56% |
| Change currency | POST | 2514 | 9% |

For stress testing the iXen application, the load generator service simulates 100 users applying 5520 requests with a rate of around 10 RPS. Table 6 shows the distribution of requests by type. Each simulated user initially logs into the application and obtains a cookie which we store and use for authentication in all requests. We have pre-configured the sensors and mashup applications to which developers and users can subscribe. In addition to simulating user-performed actions, we simulate a sensor sending random measurements.

**Table 6.** Stress-testing requests for iXen.

| Request | Request Type | Number of Requests | Distribution |
|---|---|---|---|
| Visit Homepage | GET | 957 | 17% |
| Search Available Sensors | POST | 632 | 11% |
| Subscribe Developer to Sensor | POST | 545 | 10% |
| Search Applications | POST | 639 | 12% |
| Search Application Subscriptions | GET | 654 | 12% |
| Search Subscriptions to Sensors | GET | 580 | 11% |
| Send Measurement to Sensor | POST | 604 | 11% |
| Subscribe to Application | POST | 264 | 5% |
| Deploy a Mashup Application | POST | 91 | 2% |
| Access Mashup Application | GET | 545 | 10% |
| Login into the App | POST | 100 | 2% |

### 5.5. Application Graph

Figure 6 illustrates the microservices graph of the eShop application. It is a directed acyclic graph (DAG) constructed based on information provided by the Kiali API and performance metrics extracted from the Prometheus server. It represents the placement of the default Kubernetes Scheduler on four nodes (i.e., pods with the same color are placed in the same node). HTTP and gRPC traffic between pods is measured in RPS, while TCP traffic is represented as measured in BPS.

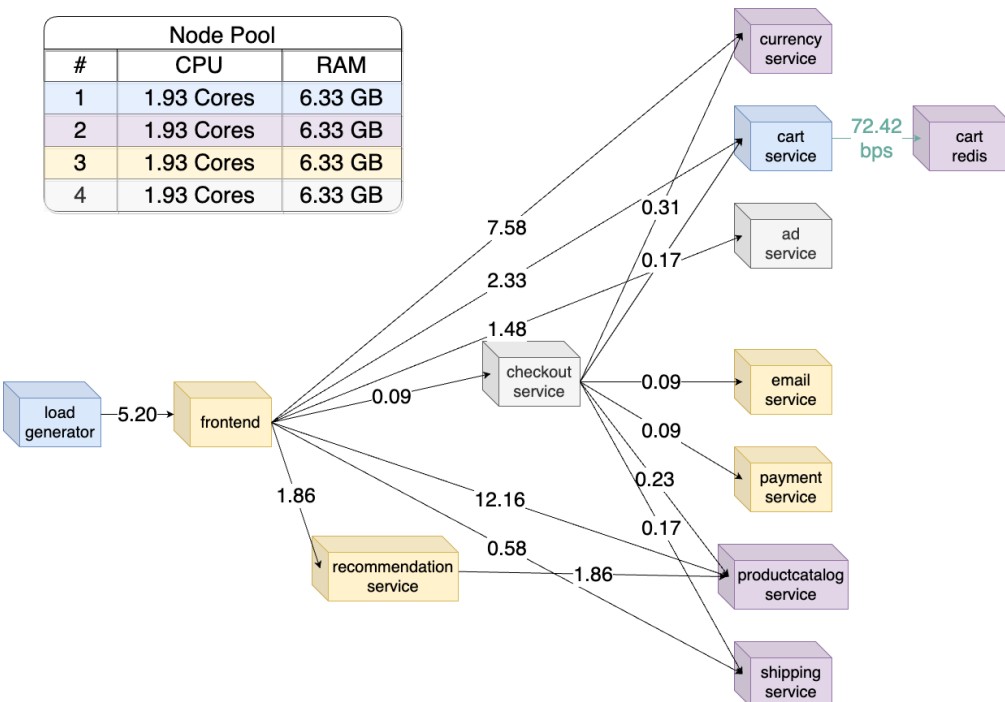

**Figure 6.** Kiali (microservices) graph of eShop application.

*5.6. Kubernetes Testbed*

We tested four different placement strategies. The first is placement by the default *Kubernetes Scheduler* of GKE. Each time an application is deployed in Kubernetes (without specifying the pod relations or the nodes that each pod must be placed in) the Kubernetes Scheduler produces a placement, which mainly depends on available node resources and the resources required by the pods. Second is placement using the *BKM-HP* method [5]. The method applies *Bisecting K-Means* hard clustering followed by *heuristic packing* (HP). The method partitions the application graph into $K = 4$ clusters and applies HP to find a placement with fewer nodes. The third method is *Heuristic First Fit* (HFF) [6], which minimizes inter-node traffic (i.e., egress network traffic) and reserves as few nodes as possible in one step. The last is *ModSoft-HP*, the proposed fuzzy service placement produced by *ModSoft* fuzzy clustering followed by HP.

Figure 7 illustrates the placement of services of the eShop application by the default Kubernetes Scheduler on four nodes. Blue lines denote egress service-to-service traffic, while gray lines denote ingress traffic. Figure 8 illustrates the placement results of the fuzzy method on three nodes and the horizontal pod autoscaler (HPA) is disabled. The fuzzy method reduced the number of nodes to three and created a replica of the *frontend* service in a different node. This service communicates with seven other services and by creating a replica almost all of its communication converts to ingress. The *productcatalog* service, which receives a lot of requests, is replicated in all three nodes to reduce the load of each replica and to achieve faster response times (i.e., if there is a destination pod instance running on the same node as the source pod, the request will be served on the same node).

Figure 9 illustrates the placement results of the fuzzy method with the horizontal pod autoscaler (HPA) enabled. ModSoft-HP retained the number of nodes as three. Compared to Figure 8, both the upper-left and bottom nodes feature two additional pods installed. The upper-left and bottom nodes have copies of the recommendation and currency services (pods). Also, copies of the frontend and recommendation services are placed in the bottom node.

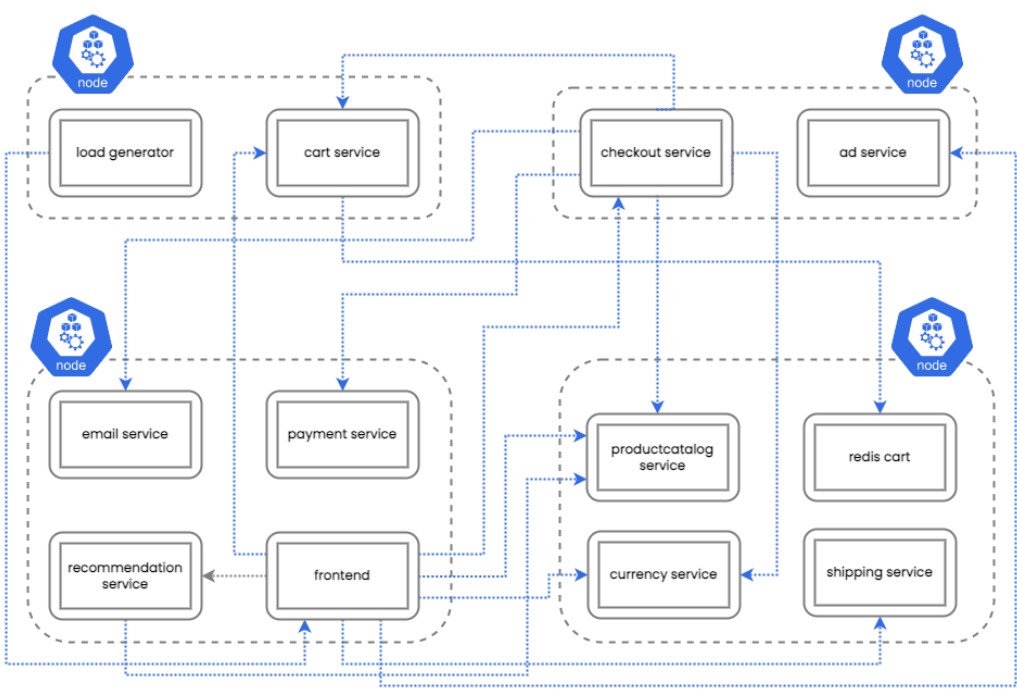

**Figure 7.** Microservices placement by the default Kubernetes Scheduler (eShop).

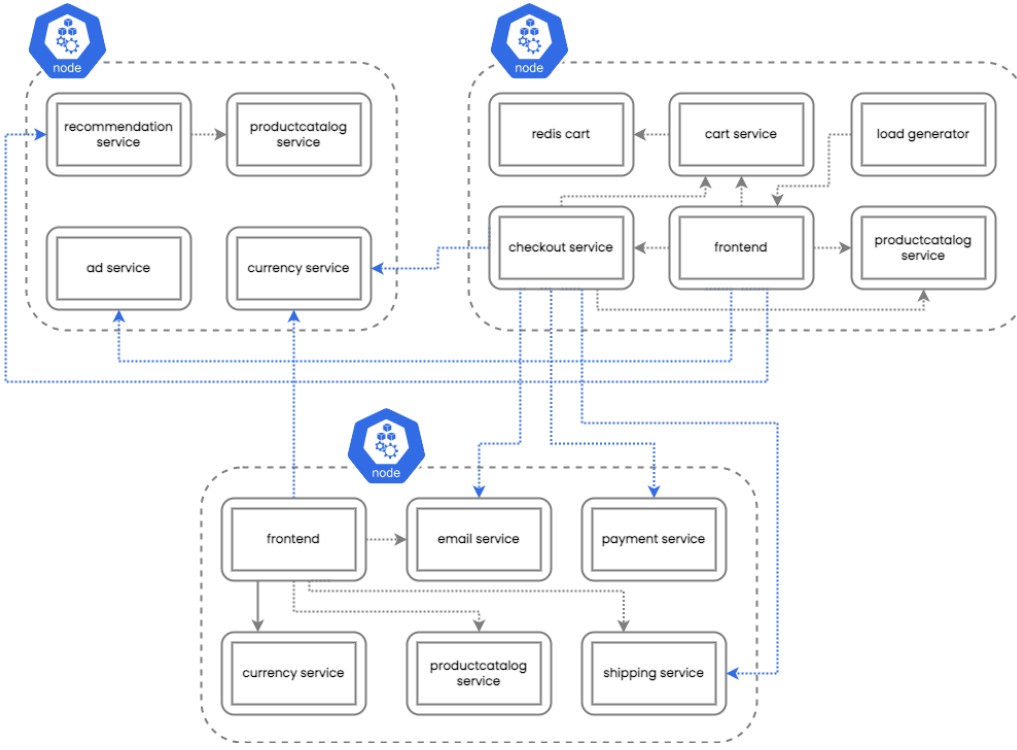

**Figure 8.** Fuzzy microservices placement with HPA disabled (eShop).

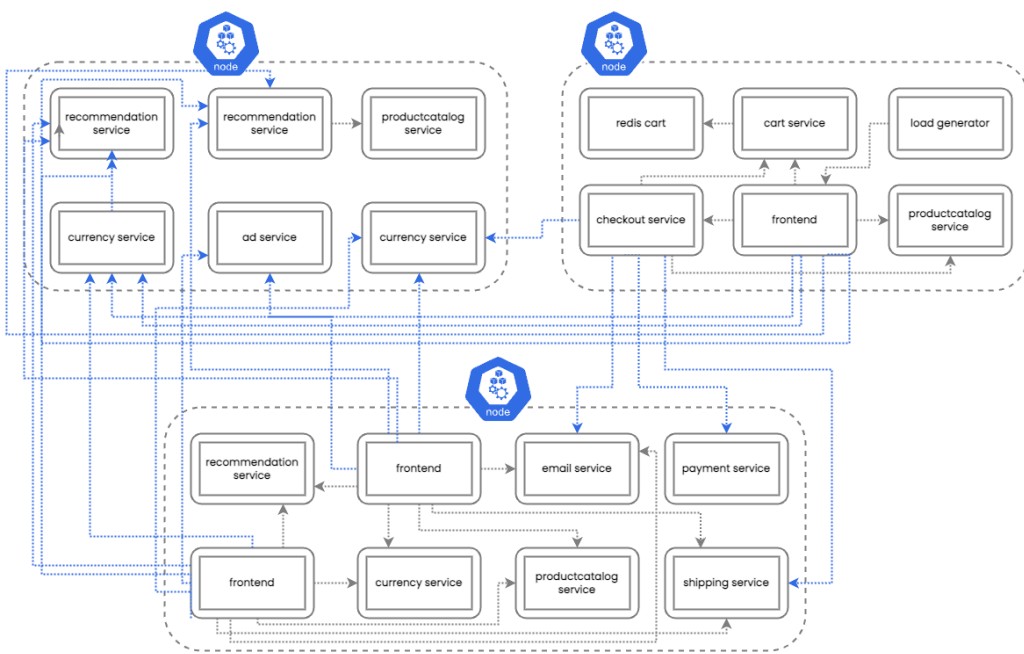

**Figure 9.** Fuzzy microservices placement with HPA enabled (eShop).

## 6. Experiments

The purpose of the following set of experiments is twofold: The first objective is to produce a fuzzy placement that reduces the total hosting cost of the infrastructure. This is achieved by reducing the number of nodes (VMs) required to host the applications and by reducing egress traffic. The second objective is to optimize the applications' response times. In the following, the results of our *fuzzy service placement strategy* (ModSoft-HP) are compared against similar results obtained by placement with the Kubernetes Scheduler and the BKM-HP [5] and HFF [6] hard clustering methods. We report results for the number of hosts used, the egress traffic, the latency (i.e., average response time over many thousands of service requests), and the monthly infrastructure cost for each method.

Service placement is determined based on the partitioning of the services graph. Graph properties are measured based on pod resource metrics and affinity metrics. As discussed in Section 5.1, WBA and RPS are alternative ways to calculate affinities between communicating services and adding weights to the edges of the services graph. As a result, graph clustering (and consequently service placement) depends on the selection of affinity metric. To study the impact of relevance on service placement, the entire set of experiments is repeated for both WBA and RPS. The experiments reveal that WBA offers a more realistic measure of the actual communication load and outperforms placement determined based on RPS in all cases.

Similar to fuzzy clustering, the HPA allows pods to be replicated (on the same or different nodes) if stressed at runtime. To study the impact of the HPA, the entire set of experiments is repeated once with the HPA disabled and once more with the HPA enabled. The experiments reveal that ModSoft-HP achieves the best performance when the HPA is enabled (i.e., outperforms all hard clustering solutions with the HPA enabled as well). Also, ModSoft-HP outperforms the K8s Scheduler even with the HPA enabled. Fuzzy clustering offers a better initial service placement solution that the HPA will adapt to workload needs.

### 6.1. Execution Time

The execution time is measured after the graph is created and the affinity metrics are produced. It stands for the time a placement method takes to execute and produce a placement solution. All methods run on a desktop machine with 3 GHz processor power, 6 cores, and 16 GB RAM. Figures 10 and 11 present the execution time of each method. The execution time for the fuzzy placement is greater than the time of BKM-HP or HFF. The fuzzy strategy produces more partitions and more services, which are given as input to the HP algorithm. All methods have acceptable execution times (i.e., a few milliseconds). These times are added to the time of the Kubernetes Scheduler.

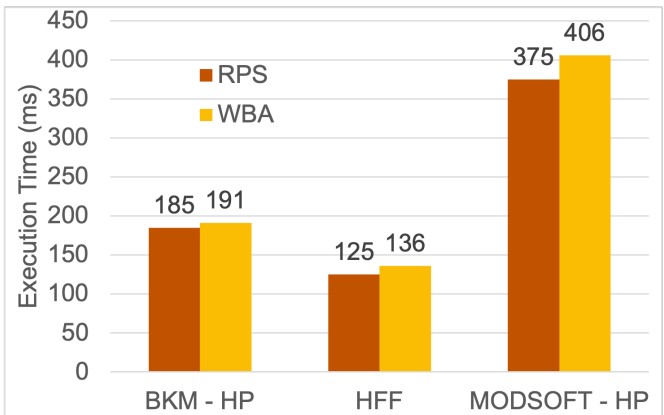

**Figure 10.** Service placement execution time for the eShop application.

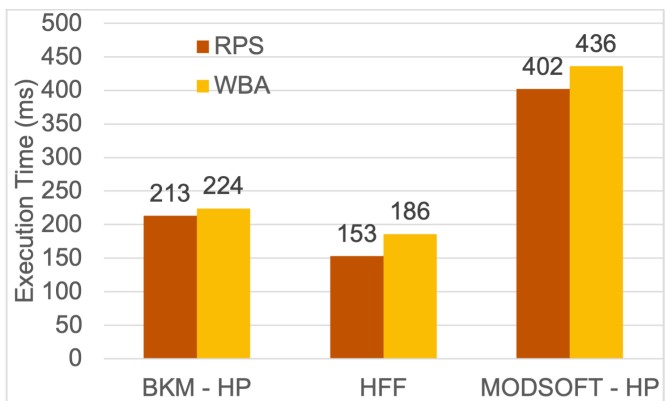

**Figure 11.** Service placement execution time for the iXen application.

### 6.2. Number of Hosts

According to Equation (7), the total cost of the infrastructure depends mainly on the number of utilized nodes ($n$). Figures 12 and 13 illustrate the number of hosts utilized for each affinity and from each generated placement. In general, the ModSoft-HP placement might require more hosts than other clustering solutions because the fuzzy placement will always produce more pods. For eShop (with fewer services), ModSoft-HP reduced the nodes to three. For iXen, all placement solutions will use four nodes when the HPA is enabled. However, the HPA does not utilize more nodes (i.e., the additional pods still fit in the same number of nodes).

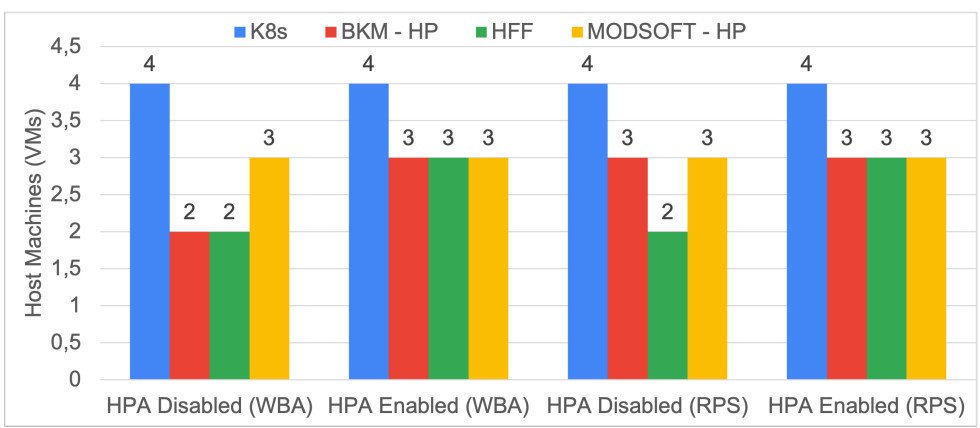

**Figure 12.** Host utilization by eShop application.

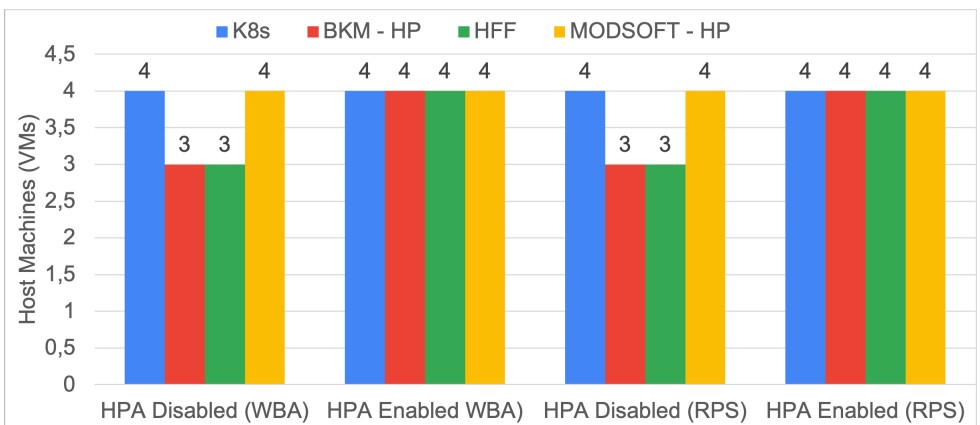

**Figure 13.** Host utilization by iXen application.

### 6.3. Egress Traffic

The message size of a request between two pods is retrieved from Prometheus. According to Google [43], to estimate the egress traffic, only the requested message size between services on different nodes is considered. Figures 14 and 15 illustrate the requested MBs between services in different nodes per hour. The fuzzy placement policy exhibited a significant reduction (i.e., up to 1/10th) in the egress traffic per hour for both applications compared to the default Kubernetes Scheduler. The selection of the affinity metric (i.e., RPS, WBA), the same as that for the HPA, had no significant impact on egress traffic.

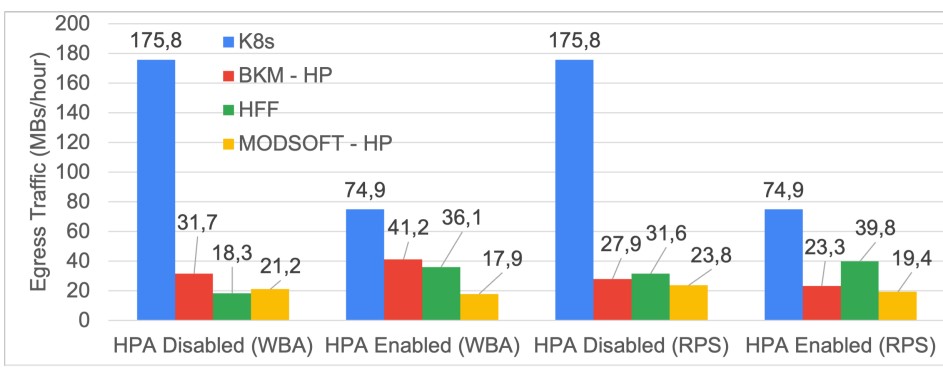

**Figure 14.** Hourly requested MBs by the eShop application.

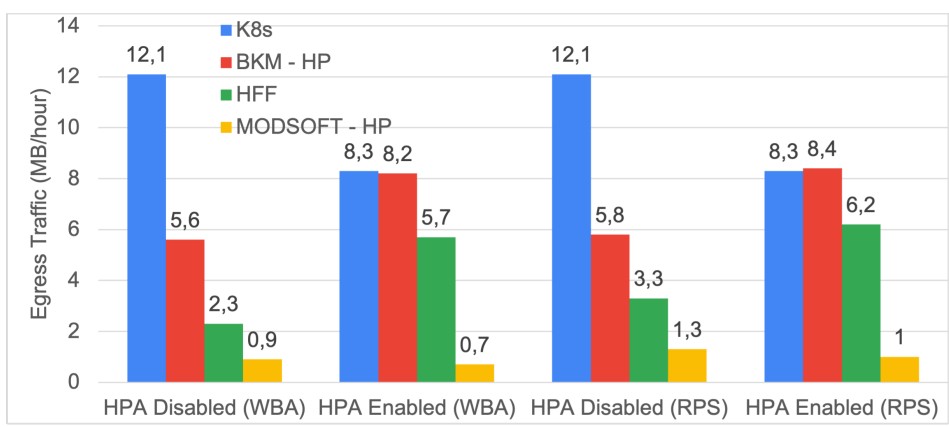

**Figure 15.** Hourly requested MBs by the iXen application.

### 6.4. Infrastructure Hosting Cost

The monthly cost for each examined placement is calculated according to Equation (7). The cost depends mainly on the number of pods (VMs) used to host each placement. The egress traffic (in GBs) also affects the total cost but not as much as the number of pods. Figures 16 and 17 illustrate the projected monthly cost for each placement using the two affinity metrics. For eShop, the difference between the BKM-HP and the ModSoft-HP placement is negligible because both placements utilize the same number of hosts. For eShop, all clustering methods managed to reduce the cost by at least 25% compared to placement with the default Kubernetes Scheduler. For iXen, no cost difference is noticeable because iXen has more microservices than eShop and their placement requires more nodes. The selection of the affinity metric or the HPA had no significant impact on cost.

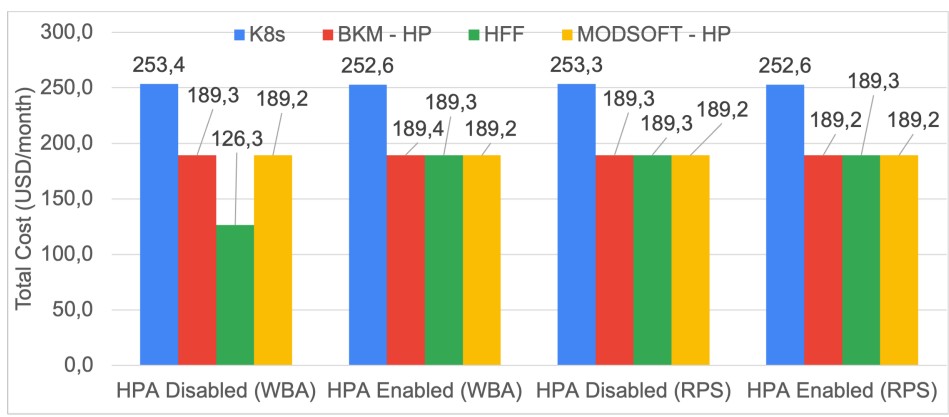

**Figure 16.** Estimated monthly infrastructure hosting cost for the eShop application.

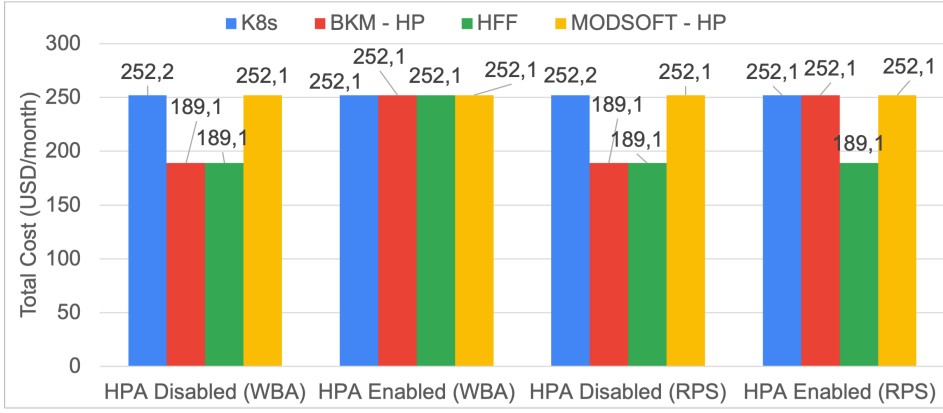

**Figure 17.** Estimated monthly infrastructure hosting cost for the iXen application.

### 6.5. Response Time

ModSoft-HP with the HPA enabled and clustering with WBA has a noticeable impact on the response time. Figures 18 and 19 report average response times (over all requests) during the stress testing. Some high-traffic services run in more than one instance and sometimes within the same pod; hence, the requests are processed faster, while each instance's load is reduced. For example, the *frontend* service, which receives most of the requests for each application, is replicated in more than one instance. The requests are balanced between these instances and are forwarded faster to their respective targets. The speed improvement with ModSoft-HP is consistently above 8% for clustering with WBA and the HPA enabled. The HPA, if enabled, improved the latency of all methods. If disabled, the speed improvement with ModSoft-HP reaches 17% for the iXen application.

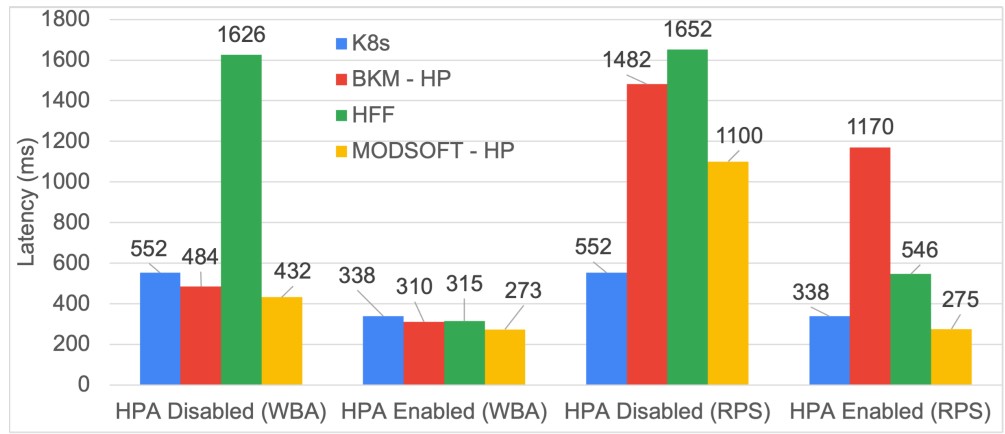

**Figure 18.** Average response time of service request for the eShop application.

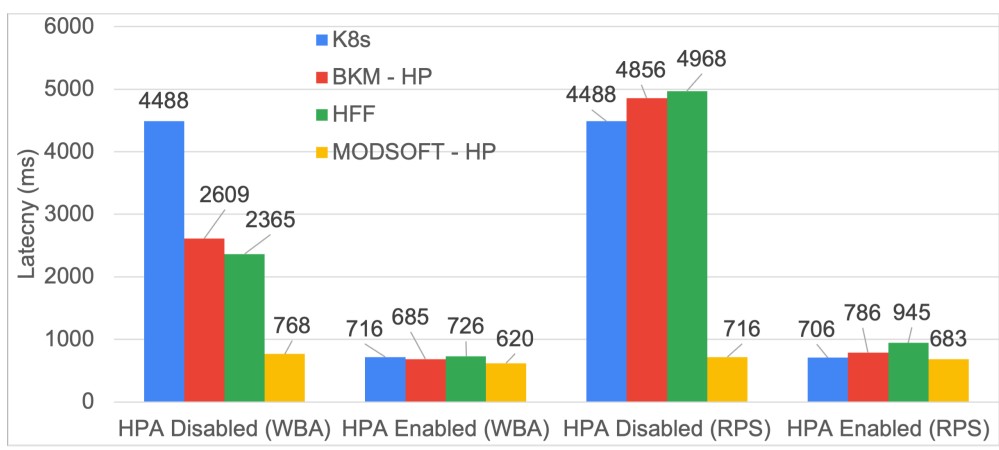

**Figure 19.** Average response time of service request for the iXen application.

### 7. Conclusions and Future Work

We investigate the problem of the optimal placement of SOA applications in Kubernetes. The decision of which services (pods) should be grouped together and run in the same node has a certain impact on the performance and hosting cost of the application. The services must be placed in nodes (VMs) in a way that minimizes a cost objective (i.e., latency, number of hosts, infrastructure hosting cost, egress traffic) or all of them combined.

The problem of service placement is formulated as a graph clustering (or graph partitioning) one. Application services form directed acyclic graphs (DAGs) with nodes representing services and edges representing communicating microservices. Both nodes and edges are labeled by the resources consumed by the services (i.e., mainly CPU and RAM for microservices) and by the affinities between them (i.e., network traffic). Graph partitioning suggests a mini-

mum set of weakly connected clusters of nodes each comprising services linked heavily with each other. This guides the placement of service clusters to Kubernetes nodes. ModSoft-HP is a fuzzy clustering method that has been embedded into a custom Kubernetes Scheduler referred to (in this work) as the ModSoft-HP Scheduler. The ModSoft-HP Scheduler allows multiple instances of an application's microservices to run on different nodes.

Weighted bidirectional affinity (WBA) and requests per second (RPS) are alternative ways to calculate affinities between communicating services and adding weights to the edges of the services graph. As a result, graph clustering (and consequently service placement) depends on the selection of the affinity metric. The experiments reveal that WBA offers a more realistic measure of the actual communication load and outperforms placement determined based on RPS in all cases. This is reasonable since WBA affinity exploits the size of the messages in partitioning the graph. The experimental results are obtained on two benchmark applications, Google's Online Boutique eShop and the iXen application for IoT data processing. ModSoft-HP with the HPA enabled and clustering with WBA has a noticeable impact on the response time. It achieves the best performance without impacting operation costs or other performance metrics. The egress traffic reduction is impressive, reaching 90% in some cases compared to deployments with the default Kubernetes Scheduler.

Although the reduction of egress traffic does not affect the monetary cost, it is expected to lead to a significant cost reduction in a heterogeneous environment (with Kubernetes nodes deployed in different regions). ModSoft-HP can prove even more effective in fog and edge cloud environments, where latency between services is significant (left as future work). In a fog-edge environment, the egress traffic reduction is expected to result in a significant reduction in hosting cost and also in latency.

**Author Contributions:** Conceptualization, E.G.M.P., A.A. and K.T.; Methodology, E.G.M.P., V.S., P.E., A.A. and K.T.; Software, V.S., P.E., A.A. and K.T.; Validation, E.G.M.P., V.S., P.E., A.A. and K.T.; Formal analysis, E.G.M.P., V.S., A.A. and K.T.; Investigation, E.G.M.P., V.S., P.E., A.A. and K.T.; Resources, V.S., A.A. and K.T.; Data curation, V.S., P.E., A.A. and K.T.; Writing—original draft, V.S., P.E., A.A. and K.T.; Writing—review & editing, E.G.M.P.; Supervision, E.G.M.P.; Project administration, E.G.M.P.; Funding acquisition, E.G.M.P. All authors have read and agreed to the published version of the manuscript.

**Funding:** This research received no external funding.

**Data Availability Statement:** Derived data supporting the findings of this study are available from the corresponding author on request.

**Acknowledgments:** We are grateful to Google for the Google Cloud Platform Education Grants program.

**Conflicts of Interest:** The authors declare no conflict of interest.

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
