# Peer review of "ModSoft-HP: Fuzzy Microservices Placement in Kubernetes"

_electronics, doi:10.3390/electronics13010065_

Round 1

Reviewer 1 Report

Comments and Suggestions for Authors

Summary:

The growing popularity of microservices architectures generated the need for tools that orchestrate their deployment in Containerized infrastructures, such as Kubernetes. Microservices running in separate Containers are packed in Pods and placed in Virtual Machines, i.e. Nodes. For applications with multiple communicating microservices, the decision of which services should be placed in the same Node has a certain impact on both the running time and the operation cost of an application. The default Kubernetes scheduler is not optimal in that case.

This work proposed graph clustering to treat the service placement problem. This work advocates that graph clustering should be fuzzy to allow high-utilized microservices to run in more than one instance, i.e., Pods, in different Nodes.

An application is modeled using a graph with nodes and edges representing communicating microservices. Graph clustering partitions the graph into clusters of microservices with high-affinity rates. Then, the microservices of each cluster are placed in the same Kubernetes Node. A class of methods resorts to hard clustering, i.e., each microservice is placed in exactly one Node.

MODularity SOFT clustering followed by Heuristic Packing (HP) (ModSoft-HP) is a Scheduler for Service Oriented Architecture (SOA) applications that takes scheduling decisions based on the results of a fuzzy clustering method.

For proof of concept:

The workloads of two applications, i.e., an e-commerce eShop and an IoT architecture, are given as input to:

1-The default Kubernetes (K8s) cluster Scheduler,

2-The Bisecting K-means (BKM) hard clustering followed by Heuristic Packing (HP) scheduler,

3-The Heuristic First Fit (HFF) clustering scheduler,

4-ModSoft-HP.

The experimental results demonstrate that ModSoft-HP can achieve up to 90% reduction of Egress traffic, up to 20% savings in response time, and up to 25% less hosting costs compared to service placement with the default Kubernetes Scheduler on Google Kubernetes Engine.

Strengths:

Important problem

Good idea

Good experiments

Good explanation

Weaknesses:

1-Lines 13 and 67, (ModSoft-HP) will be changed to (MODularity SOFT (ModSoft) clustering followed by Heuristic Packing (HP) (ModSoft-HP) [37])

2-From the references [7][31][36], the idea of fuzzy graph clustering is relatively old.  Can you explain why the idea of using fuzzy graph clustering to treat the service placement problem did not appear earlier?

3-Line 371, add a paragraph that explains the relation between Algorithm 1 and Algorithm 2.  Make it clear when you apply them to your experiments.

4-Some references need to be written with details, for example references [29][40][41][42][43][44]…..etc

5-Please can you support us with a place at github, for example, that contains data, results, programs, code, simulation, and other materials used in your research.

Author Response

We thank the reviewer for his/her insightful comments. Please find below answers to the points of the review (please notice that the order of references has changed in the revised text).

Comment 1: “1-Lines 13 and 67, (ModSoft-HP) will be changed to (MODularity SOFT (ModSoft) clustering followed by Heuristic Packing (HP) (ModSoft-HP) [37])”

Action: The roles of Soft Clustering (ModSoft) algorithm and the ModSoft-HP Scheduler are not clear in the text. We thank the reviewer for pointing this out.

ModSoft followed by Heuristic Packing is a fuzzy clustering method that decides where application services must be placed using minimum resources. The ModSoft-HO Scheduler is a custom K8s Scheduler that uses the results of ModSoft-HP fuzzy algorithm (invokes a clustering solution in the preamble) to instruct the scheduler where to place each service.

We reworded lines 1 and 13 and added comments throughout the text to clarify the issue.

Comment 2: “2-From the references [7][31][36], the idea of fuzzy graph clustering is relatively old.  Can you explain why the idea of using fuzzy graph clustering to treat the service placement problem did not appear earlier?”

Action: Although the idea of fuzzy clustering is old (as far as we know) it has not been exploited for the service placement problem. However, Soft Modularity optimization is a relatively new concept, and its polynomial time implementation is not old [38, 8]. This is discussed in Section 4.1 (first paragraph). Therefore, ModSoft-HP Scheduler exploits recent (rather than old) research results on fuzzy clustering to improve service placement in GKE.

Comment 3: “3-Line 371, add a paragraph that explains the relation between Algorithm 1 and Algorithm 2.  Make it clear when you apply them to your experiments.”

Action: See also the answer to comment 1. The relation between the two algorithms is clarified in Section 4.2 (second paragraph). ModSoft-HP fuzzy clustering is embedded into a new custom Kubernetes scheduler. ModSoft-HP clustering decides where application services must be placed. The ModSoft-HP Scheduler invokes ModSoft-HP clustering (i.e. line 9 of Algorithm 2) to instruct the Scheduler where (i.e., to which Node) to place each service.

Comment 4: “5-Please can you support us with a place at github, for example, that contains data, results, programs, code, simulation, and other materials used in your research”

Action: ModSoft-HP Scheduler and the SOA placement problem as described in the paper is a work in progress and is part of the thesis of the authors. We would hesitate to hand everything out until the research is completed.

Reviewer 2 Report

Comments and Suggestions for Authors

This article presents a fuzzy-based service placement method in Kubernetes. It proposes ModSoft-HP scheduler that combines modularity optimization and heuristic packing methods. The experimental environment and target (real-world) applications are clearly demonstrated. The experimental evaluation, as well, is presented in detail with various metrics such as execution time, the number of hosts, egress traffic, hosting costs, and so on. Those are of great value.

The proposed method and implementation details however should be extended by providing more details. The author proposes a new idea called ModSoft-HP, however, they just have cited their technical report [39] without detailed explanation. Algorithm 2 is not enough. Furthermore, authors should clearly distinguish what is proposed in this paper from what is not proposed; ModSoft algorithm described in Sec 4.1 seems not to be contribution of this paper, but it is described as if it were a proposed technique. It is strongly recommended to reorganize the paper to distinguish the background and the proposed technique. 

Comments on the Quality of English Language

Extensive proofreading is required (e.g., line 277: "It The~", Table 1: "fyzzy", ...).

Author Response

We thank the reviewer for his/her insightful comments. Please find below answers to the points of the review (please notice that the order of references has changed in the revised text).

Comment 1: “The author proposes a new idea called ModSoft-HP, however, they just have cited their technical report [39] without detailed explanation. Algorithm 2 is not enough.”

Action: Section 4.2 and Algorithm 2 provide an overview of the custom Kubernetes scheduler proposed in this work. All key details are included in the text. Information about Kubernetes scheduler customization is available in [1], [2] and [3]. References are added at the end of Section 4.2.

Application is purely a matter of technical skill, not research. The related discussion will raise more questions that would require more clarification about terminology and Kubernetes. This discussion would be outside the scope of the paper. Reference [39] refers to the author's thesis available online.

See also answers to the first reviewer's comments.

Comment 2: “Furthermore, authors should clearly distinguish what is proposed in this paper from what is not proposed; ModSoft algorithm described in Sec 4.1 seems not to be the contribution of this paper, but it is described as if it were a proposed technique. It is strongly recommended to reorganize the paper to distinguish the background and the proposed technique. “

Action: With all due respect to the reviewer, it is clear from the discussion in Section 4.1 that soft modularity clustering is not a contribution. However, Algorithm 1 is an integral part of the approach. Only essential details are described in Section 4.1. We choose to include Algorithm 1 in Section 4.1 to streamline the presentation of ModSoft-HP. Otherwise, the description of ModSoft-HP would be split between Section 2 and Section 4, and it would be more difficult for the reader to follow the discussion.

Round 2

Reviewer 2 Report

Comments and Suggestions for Authors The authors' clarification has resolved my concerns in a satisfactory manner. In my eyes, it is good to be accepted.